## Research Article

net ecosystem carbon exchange (NEE); stand transpiration; lateral fluxes; nitrogen; phosphorus

**Corresponding author:**
Ken W. Krauss;
Email: kkrauss@lumcon.edu

# Excessive phosphorus loading contributes to future vulnerability of mangrove ecosystems by reducing net ecosystem exchange of carbon

Ken W. Krauss[1] , Jeremy R. Conrad[2], Jamie A. Duberstein[3], Eric J. Ward[4], Judith Z. Drexler[5], Kevin J. Buffington[6], Brian W. Benscoter[7], Haley Jane Miller[3], Natalie T. Faron[8], Sergio Merino[9], Andrew S. From[9], Elista Peneva-Reed[10], Zhiliang Zhu[10], Karen M. Thorne[6] and Ilka C. Feller[11]

[1]Louisiana Universities Marine Consortium, USA; [2]Inventory and Monitoring Program, US Fish and Wildlife Service Southeast Region, USA; [3]Baruch Institute of Coastal Ecology and Forest Science, Clemson University, USA; [4]University of Maryland, USA; [5]California Water Science Center, US Geological Survey California-Great Basin Region, USA; [6]Western Ecological Research Center, US Geological Survey California-Great Basin Region, USA; [7]Department of Biological Sciences, Florida Atlantic University, USA; [8]Florida Atlantic University, USA; [9]Wetland and Aquatic Research Center, US Geological Survey Southeast Region, USA; [10]US Geological Survey, USA and [11]Smithsonian Environmental Research Center, Smithsonian Institution, USA

## Abstract

J.N. "Ding" Darling National Wildlife Refuge (DDNWR) is located on Sanibel Island along the southwestern coast of Florida, USA. There, eutrophication attributed to agricultural discharge along the Caloosahatchee River has affected the area's aquatic habitat. In anticipation of additional nutrient loading, we experimentally fertilized mangrove forests with nitrogen (+N; NH4) and phosphorus (+P; P2O5) for 3 years, and monitored soil and pneumatophore CO2 fluxes and tree sap flow from two mangrove species. Furthermore, we modeled individual tree and stand water use, from which we developed carbon (C) budgets for +N and + P vs. control simulations based on a novel application of water use efficiency conversion. Many of the measured response variables provided hints of subtle changes in response to +P rather than +N, which were enhanced when scaled. From this, we found that additional P loading is expected to reduce both gross and net primary productivity as well as CO2 uptake via net ecosystem exchange of C, likely pressing the system beyond metabolic capacity and leading to a 48–62% decrease in projected lateral C export. Greater eutrophication will likely compound vulnerabilities to sea-level rise submergence, especially where P concentrations are high and already reducing soil surface elevations.

## Impact statement

After a hurricane in 2004, recovery of the mangroves on Sanibel Island was slow versus previous hurricanes. Along with more development around the mangroves, what also changed was the amount of nutrient loading into the mangroves from the Everglades Agricultural Area. Soil total P was three to four times higher on Sanibel Island than in other Florida mangroves, while soil total N was not distinctive. Our goal was to learn how net ecosystem exchange of carbon might be affected by additional N and P loading expected as a future condition. We developed a plan to use leaf-scale water use efficiencies versus sap flow-derived stand water use to determine gross primary productivity (GPP). The components of the carbon budget cascade from GPP determination, and we included additional measurements of soil carbon burial as well as component canopy, soil and pneumatophore respiration. We discovered that additional P loading can force loss of carbon uptake in lieu of the gains documented for control and N-fertilized plots. Whether this reduction in carbon uptake, and subsequent export is sustained over time is not known, but preventing additional nutrient loading to Sanibel's mangroves weighs heavily.

## Introduction

Threats to mangrove ecosystems are manifold. Mangroves are important areas targeted by agricultural development to produce sustaining crops such as rice (Richards and Friess, 2016). Mariculture and forestry practices (e.g., oil palm plantations) have diversified the world's agricultural economy and thus extended the global probability of further mangrove habitat decline (Goldberg et al., 2020). For decades, the call to slow mangrove area losses has resonated with scientists (Duke et al., 2007), but less so with policies intended to drive economic development of specific nations (Golebie et al., 2022). Indeed, mangrove habitat

destruction remains a global concern, not only through land conversion but also through subtle stressors perhaps not gaining immediate notice. Through education, conservation, restoration and latitudinal expansion, mangrove area losses have slowed considerably in recent years (Feller et al., 2017), but losses still remain large in some places (Friess et al., 2020). Furthermore, these statistics do not incorporate a decline in mangrove health as a loss of function, related to critical valuation provided by carbon (C) sequestration, water quality improvement, sediment filtering and fisheries habitat (Friess et al., 2019; Wang et al., 2020; Xu et al., 2024). Degradation is often subtle and unnoticed.

Two threats to mangrove habitat degradation stand out. The first, is hydrological disturbance. Mangroves develop as intertidal habitat, and an acceptable balance of osmotic, ionic and phytotoxin tolerances relate strongly to frequent exposure of soils to the atmosphere (i.e., most mangroves are inundated for <30% of the year, Lewis, 2005), escape from hypersalinity and flushing of hydrogen sulfide (Koch et al., 1990; Behera et al., 2015). Road construction, causeway projects, mosquito impoundment, inadequate culverting and erosion control (e.g., sea walls) often cause disturbance to hydroperiod (Lewis et al., 2016).

A second threat is eutrophication (Reef et al., 2010). The relationship among nitrogen (N), phosphorus (P) and mangrove growth has generally established that P added to mangrove soils on carbonate settings leads to greater growth potential and that mangrove species, such as *Rhizophora mangle*, produce morphological attributes such as sclerophyllous leaves in low-nutrient environments to promote greater nutrient retention (Feller, 1995). While growth enhancements may appear adaptive, a global analysis revealed that high levels of anthropogenic eutrophication result in greater mangrove mortality as trees allocate more resources to shoot biomass versus root biomass (Lovelock et al., 2009). Root growth is key to mangrove survival and elevation maintenance (McKee, 2011). Furthermore, nutrient enrichment increases soil respiration in some settings (Lovelock et al., 2014a), although hydroperiod presents an important confounding variable (McKee et al., 2002). The potential for processing additional nutrients relative to the metabolic limits of mangroves is unreconciled for many of south Florida's estuaries.

Understanding the future fate of mangrove C with eutrophication in a south Florida estuary is the objective of this study. We focus on J.N. "Ding" Darling National Wildlife Refuge (DDNWR) located on Sanibel Island adjacent to Ft. Myers, Florida, USA. Through a series of studies targeting soil C storage, soil and aerial root $CO_2$ flux, tree sap flow and leaf-scale water use efficiencies, we describe the potential influence that increased inputs of nitrogen (N) and phosphorus (P) might have on mangrove C fluxes. We model stand water use, estimate gross primary productivity through a relatively new approach for forested wetlands, develop a refuge-scale C budget and document how the C budget may be altered with additional N and P loading. We hypothesized that increased future loading of N and P in an already eutrophic ecosystem will result in a detrimental shift in mangrove energetics.

## Methods

### Site description

Most mangrove forests at DDNWR are located on Sanibel Island, Florida (Figure 1a) and include mixes of *R. mangle*, *Avicennia germinans* and *Laguncularia racemosa*. After a hurricane in 2004

(Hurricane Charley), Peneva-Reed et al. (2021) discovered that Sanibel Island's mangroves recovered slowly, directing concerns about stand health. Sanibel Island's mangroves are positioned near the mouth of the Caloosahatchee River, which serves as a significant flood control conveyance for the Everglades Agricultural Area (EAA), Lake Okeechobee and the Northern Everglades. This canal-river-bay system will likely contribute to even greater nutrient loading ($NO_x$, $PO_4$) in the future if actions are not taken to reduce nutrient throughput (Liu et al., 2009; Buzzelli et al., 2014). Currently, Lake Okeechobee contributes up to 32% of the nutrients found in the Caloosahatchee watershed (through mineralization of buried nutrients), non-tidal basin areas contribute 48% and tidal areas contribute the remaining 20% (Lee County, 2019).

### Experimental design

While soil N concentrations were not distinctive, soil P concentrations were three to four times higher at DDNWR than like-positioned mangroves in southwest Florida (Conrad, 2022). We designed a three-year fertilization experiment to learn more about present (control) and future increases (+N and + P amendment) in nutrient loading (Conrad et al., 2024) in basin and fringe hydrogeomorphic zones. Basin mangroves occur approximately 20 m from the edges of bays and extend inland with slow water drainage after high spring tides (salinity, 46 psu). Fringe mangroves align bays, typically occupying 20-m bands just between basin mangroves and open water (salinity, 35 psu). Our experimental design consisted of these two hydrogeomorphic zones in three locations, or blocks, positioned at least 250 m apart. Each block was oriented perpendicular to the shoreline spanning from the bay and consisted of six, 8x8 m plots (64-m$^2$), with three plots in basin mangroves and three plots in fringe mangroves. Plots were positioned a minimum of 25 m from each other for a total of six plots per block x 3 blocks, or 18 experimental plots (treatment combinations), occupying a shoreline distance of ~0.5 km (Conrad et al., 2024).

### Nutrient treatment/fertilization application

We randomly assigned plots to receive one of three nutrient treatments: +N, +P or control (no amendment). Granular nitrogen (urea $NH_4$–45:0:0, N-P-K) or phosphorus (superphosphate $P_2O_5$–0:45:0, N-P-K) was used based on previous application experience (Feller, 1995; McKee et al., 2002; Feller et al., 2003; McKee et al., 2007). Urea $CO(NH_2)_2$ readily converts to ammonium ($NH_4$), the primary form of mineralized N found in mangrove soils (Feller, 1995).

As these are tidal settings and flooded for 16% (basin) to 30% (daily tides, fringe) per year at DDNWR (Conrad, 2022), granular nutrients were buried into soils for slower release. A 2-m grid was overlain on plots, and a fertilizer hole was dug at each 2x2-m intersect, totaling 16 fertilizer holes per plot. These holes have an effective fertilization radius of up to 1 m (Feller et al., 2003) with persistence up to 6 months, which we verified through repetitive porewater sampling at DDNWR (Miller, 2022; see expanded description in Supplementary Material). Holes were augured to a 2.5 cm diameter and 30 cm depth. A fertilizer mass of 150 g was added to each hole with replacement of soil. For control plots, cores were extracted and replaced without fertilizer. Our fertilizer regimes were selected based on threshold deliveries of N and P previously discovered to elicit a growth response in *R. mangle* (Feller, 1995). Fertilizers were applied twice annually for three years (2018–2020). Plots were fertilized in summer/fall to coincide with

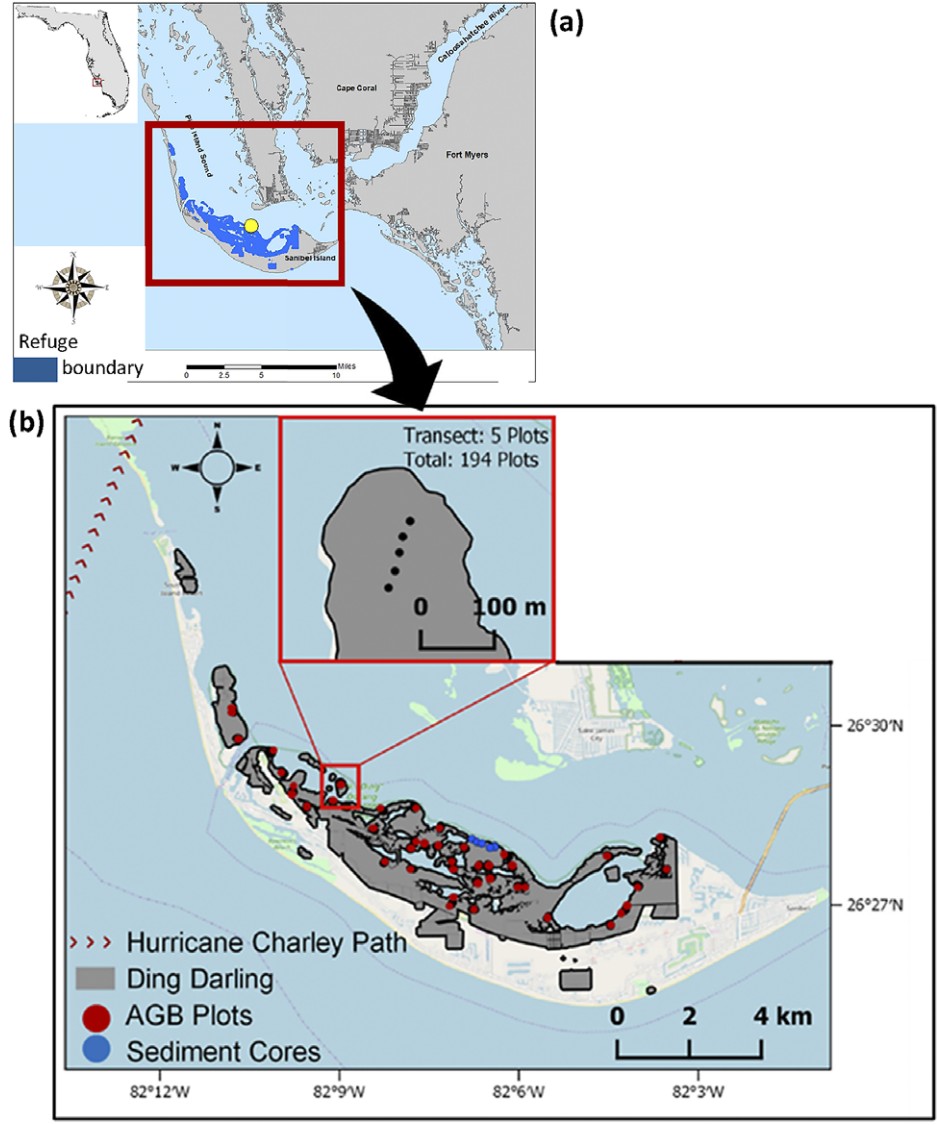

**Figure 1.** (a) Location of Ding Darling National Wildlife Refuge (DWR) in southwest Florida, USA, downstream of the Caloosahatchee River outflow and the source of nutrient loading to the waters around Sanibel Island. (b) Location of sediment cores and aboveground (AGB) forest structural plots (129 plots) sampled across the refuge.

peak Caloosahatchee River discharge and winter to match average discharge; however, river discharge depends on managed Lake Okeechobee levels and is inconsistent.

## Sap flow

We measured sap flow ($J_s$) in *A. germinans* (*dbh*, 6.1–40.9 cm) and *R. mangle* (*dbh*, 6.7–32.6 cm) trees occupying +N, +P and control plots beginning one year after first fertilization (Table 1). Raw $J_s$ data are $d$T values, which we recorded over two summers and one winter. Data from every sensor were recorded at 30-min intervals, and $d$T converted to $J_s$ rates (g $H_2O$ m$^2$ s$^{-1}$) relative to insertion depths of 5, 15, 50, 70 and 90 mm using formulas described by Granier (1987). We determined functional sapwood area through $J_s$ measurements at different radial depths (Krauss et al., 2015), with the attenuation of sap flow into each tree determined separately for

**Table 1.** Sample size and duration of acceptable monitoring records for trees selected for sap flow measurement by deloyment, species and treatment at Ding Darling NWR

| Treatment | Growing season 2019 (2 May to 15 August) | | | Dry season 2020 (3 February to 23 March) | | | Growing season 2020 (6 June to 5 October) | | |
|---|---|---|---|---|---|---|---|---|---|
| | *Rhizophora mangle* | *Avicennia germinans* | Days | *R. mangle* | *A. germinans* | days | *R. mangle* | *A. germinans* | days |
| Control | 4 | 4 | 92 | 2 | 4 | 49 | 3 | 4 | 120 |
| +N | 4 | 4 | 45 | 2 | 4 | 49 | 3 | 5 | 123 |
| +P | 6 | 6 | 104 | 6 | 6 | 49 | 6 | 6 | 122 |

control, +N and + P (Table 2). Maximum $J_s$ and radial attenuation curves for *L. racemosa* were attained from a previous study near DDNWR (~58 km) (Krauss et al., 2007) and were indicative of control treatments. Additional details are provided in the Supplementary Material.

Analyses were restricted to fair weather days to avoid confounding environmental conditions during two late spring/summer periods (Krauss et al., 2007), as follows: 6 May to 25 June 2019 (32 days used, out of a maximum of 45) and 20 June to 21 August 2020 (39 days used, out of a maximum of 120). $J_s$ data were also analyzed from the end of the northern hemisphere winter from 6 February to 22 March 2020. Modeling assumes maximum average individual tree water use by hydrogeomorphic zone, species, depth and treatment (control, +N and +P) versus environmental variables from a nearby weather station to scale to daily, weekly and annual water use (Krauss et al., 2015). Raw $J_s$ data are available in Duberstein et al. (2023).

### Stand water use

Sap flow measurements were converted to tree water use ($F$: mm y$^{-1}$) through an individual-based spreadsheet model that compartmentalizes data from each tree, size class and species-specific

maximum average $J_s$ (Krauss et al., 2015). Determinations of $F$ by hydrogeomorphic zone, species and *dbh* were applied to data collected from 129, 7-m-radius plots previously surveyed at DDNWR (Peneva-Reed et al., 2021), but by avoiding plots with heavy Hurricane Charley damage (Figure 1b). $F$ was converted to stand water use ($S$) by using a biometric model (Čermák et al., 2004; McLaughlin et al., 2012), modified for forested wetlands (Krauss et al., 2015) and updated with additional detail for area calculation, as follows:

$$S = \frac{\sum_{i=1}^{n}\left((F_{tree})_i \times \frac{10,000}{BA_i}\right)}{A} \times \frac{1}{\rho} \qquad [1]$$

where $i$ corresponds to $F$ from individual trees in a stand, 1 is the first tree in a stand over a diameter limit of ≥0.1 cm, 2 is the second tree (etc …), $BA$ is that individual tree's basal area at a height of 1.3-m, $A$ is the ground area occupied by the stand surveyed (m$^2$) with 10,000 m$^2$ framing that value in hectares, and $\rho$ is the density of water (0.998 g cm$^{-3}$). Among each of the 129 plots, basal area data were collected over a ground area of 153.86 m$^2$ for trees ≥5.0 cm *dbh* and 12.57 m$^2$ for trees ≥0.1 cm but <5.0 cm *dbh*. $S$ is reported as kg H$_2$O m$^{-2}$ ground area, which is identical to the same value in millimeters (mm). Based on a previous analysis (Krauss et al.,

**Table 2.** Maximum rates of sap flow by mangrove species at Ding Darling NWR for modeling individual tree and stand water use, and the associated attenuation multiplier used to account for radial sapwood depth

| Depth into sapwood (cm) | Sapflow (maximum, g cm$^{-2}$ s$^{-1}$) | | | | | |
| --- | --- | --- | --- | --- | --- | --- |
| | 0 to ≤1 | 1 to ≤3 | 3 to ≤6 | 6 to ≤8 | 8 to ≤10 | >10 |
| ***Rhizophora mangle*** | | | | | | |
| All treatments | 0.002487 | 0.004609 | 0.002062 | 0.002922 | 0.001412 | 0.000000 |
| *multiplier* | **0.539** | **1.000** | **0.447** | **0.634** | **0.306** | **0.000** |
| +N | 0.001783 | 0.004293 | 0.004099 | 0.002722[a] | 0.000356 | 0.000000 |
| *multiplier* | **0.415** | **1.000** | **0.955** | **0.634**[b] | **0.083** | **0.000** |
| +P | 0.003167 | 0.004725 | 0.002112[a] | 0.002996[a] | 0.001446[a] | 0.000000 |
| *multiplier* | **0.670** | **1.000** | **0.447**[b] | **0.634**[b] | **0.306**[b] | **0.000** |
| Control | 0.002496 | 0.004710 | 0.001383 | 0.002922 | 0.001765 | 0.000000 |
| *multiplier* | **0.530** | **1.000** | **0.294** | **0.620** | **0.375** | **0.000** |
| ***Avicennia germinans*** | | | | | | |
| All treatments | 0.003257 | 0.005326 | 0.002492 | 0.001528 | 0.001605 | 0.000000 |
| *multiplier* | **0.612** | **1.000** | **0.468** | **0.287** | **0.301** | **0.000** |
| +N | 0.002739 | 0.005717 | 0.001724 | 0.001096 | 0.002214 | 0.000000 |
| *multiplier* | **0.479** | **1.000** | **0.302** | **0.192** | **0.387** | **0.000** |
| +P | 0.003270 | 0.004732 | 0.003072 | 0.001111 | 0.001355 | 0.000000 |
| *multiplier* | **0.691** | **1.000** | **0.649** | **0.235** | **0.286** | **0.000** |
| Control | 0.003231 | 0.005214 | 0.002868 | 0.001949 | 0.000887 | 0.000000 |
| *multiplier* | **0.620** | **1.000** | **0.550** | **0.374** | **0.170** | **0.000** |
| ***Laguncularia racemosa*** | | | | | | |
| All treatments | 0.002598[c] | 0.004511[c] | 0.003153[c] | 0.001304[c] | 0.000568[c] | 0.000000[c] |
| *multiplier* | **0.576** | **1.000** | **0.699** | **0.289** | **0.126** | **0.000** |

[a]Not available from study (small sample size); used multiplier from "All treatments."
[b]Not available from study (small sample size); value from "All treatments" used.
[c]Data from Krauss et al. (2007); no adjustments made for +N and +P.

2015), we estimate that calculations applied to >60 plots would be required to differentiate among treatments that differ in $S$ by 100 mm y$^{-1}$.

### Carbon fluxes

#### Soil carbon burial and soil surface C accretion

Prior to fertilization, sediment cores were taken to a depth of 1-m, sectioned into 2-cm segments to 50-cm, sectioned into 10-cm sections to 100-cm and analyzed for bulk density (g cm$^{-3}$) and organic carbon content (%) (Drexler, 2019). A rod surface elevation table (rSET) was installed to a refusal depth of approximately 8.0 ± 0.5 m (mean ± SE) in each plot ($N$ = 18), enabling repetitive measurements of soil surface elevation change over the 3 years of fertilization (Conrad et al., 2024). Soil C burial was determined over the application period as surface elevation change relative to the C content of the upper 2-cm of soil (Lovelock et al., 2014b). Additionally, using the same C content, we used marker horizons of feldspar clay (Cahoon and Lynch, 1997), three per plot ($N$ = 54), to assess soil surface C accretion, or SSCA.

#### Soil, pneumatophore and canopy losses of $CO_2$

Soil $CO_2$ fluxes ($R_s$), which include microbial soil and dead root respiration as well as live root respiration from the rhizosphere, and pneumatophore $CO_2$ fluxes ($R_p$), which were the prominent aerial root type in the mangrove forests of DDNWR, were measured using portable infrared gas analyzers over 4 days each in June (wet season) and November (dry season) by nutrient treatment (Faron, 2021; Supplementary Material). $CO_2$ fluxes associated with canopy respiration ($R_c$) were determined as the percentage of dark respiration ($R_d$) versus gross C assimilation from leaf gas exchange measurement. $R_d$ offset approximately 15.56%, 19.49% and 19.66% of gross C assimilation at the leaf scale for *R. mangle*, *A. germinans* and *L. racemosa*, respectively (Krauss et al., 2006; Chieppa et al., 2023).

#### Gross primary productivity, net primary productivity and net ecosystem exchange

We determined gross primary productivity (GPP) through a recently developed STrAP (Stand Transpiration and Productivity) model, which estimates the total amount of C taken up by the mangrove canopy in the form of $CO_2$ based upon the amount of water the mangrove canopy used in 2019 and 2020. For this model, we measured WUE$_i$ by species and nutrient treatment in-situ using an infrared gas analyzer (model Li-6,800, Li-Cor Environmental, Inc., Lincoln, NE, USA) during a mid-summer campaign in August of 2020 on new fully extended *A. germinans* and *R. mangle* leaves developing in control, +N and +P plots. We held humidity constant and used a light saturated PPFD of 1800 μmol m$^{-2}$ s$^{-1}$ (Krauss et al., 2006). Given the small numbers of *L. racemosa* on plots, WUE$_i$ for *L. racemosa* was extracted from the literature and applied consistently across treatments (Krauss et al., 2006).

WUE$_i$ adjustments were applied to annual values of $S$ by species; $S \propto$ GPP by the relationship of WUE$_i$ (Liang et al., 2022). Because each of the 129, 7-m-radius plots had a different basal area distribution by species (Peneva-Reed et al., 2021), we weighted estimates of $S$ by species during calculations of GPP using the proportional contribution of individual species to $S$ for each plot (from Peneva-Reed and Zhu, 2019). GPP was converted to net primary productivity (NPP) by subtracting $R_c$.

### Estimating lateral flux and net ecosystem carbon balance

Carbon fluxes into the mangroves are denoted as (−) and carbon fluxes out of the mangroves as (+). Using this contention, we estimated lateral C fluxes as follows:

$$Lateral\ flux = Soil\ C\ burial - NEE \qquad [2]$$

Where soil C burial (−) denotes C storage rate using SET techniques, NEE represents GPP minus $R_c$, $R_s$ and $R_p$, denoted (in this formula) as either net uptake (−) or net loss (+), and lateral flux is calculated through differential either as export (+) or import (−). In addition to determining component C fluxes as g C m$^{-2}$ y$^{-1}$, we also scale these values primarily for discussion purposes (in Gg C y$^{-1}$) based on a mangrove area of 1,112 ± 116 ha for DDNWR (Peneva-Reed et al., 2021), comprising 68% basin mangroves and 32% fringe mangroves (J. Conrad, unpubl.).

### Analysis

One-way and two-way ANOVAs were used on either transformed data to attain normality or non-normal data (Kruskal–Wallis Rank Test) to determine differences in $J_s$ by radial depth, as well as for $S$, $R_s$ and $R_p$ by location and treatment. Linear regressions were used for $R_s$ and $R_p$ prediction. All data were analyzed statistically using the programs R (v. 4.0.5) (R Core Team, 2020) for two-way ANOVAs or SigmaPlot (v. 16) for one-way ANOVAs and regression.

### Results

### Sap flow ($J_s$)

We found no effect of treatment (control, +N, +P) on average daily maximum rates of $J_s$ in either *A. germinans* or *R. mangle* at a radial sapwood depth of 5 mm and 15 mm; however, across all treatments $J_s$ at 5 mm was only 62% and 53% of $J_s$ at 15 mm by species, respectively (Table 2). Despite this, for maximum flows during wet season periods, $J_s$ at 5 mm was higher across both species ($F_{1,1} = 6.886$, $p = 0.011$), with a least square means of 18.2 g H$_2$O m$^{-2}$ s$^{-1}$. Also, the two species differed in how they used water at specific depths into the sapwood, which varied with combinations of species and depth (Figure 2). For example, at a radial depth of 15 mm, *A. germinans* registered significantly higher ($F_{1,1} = 7.917$, $p = 0.007$) overall average daily maximum $J_s$ across all seasons, with a least square mean of 41.7 g H$_2$O m$^{-2}$ s$^{-1}$ compared to *R. mangle* (least square mean, 33.7 g H$_2$O m$^{-2}$ s$^{-1}$).

In all, three important patterns emerged that affected the way water was used by individual species relative to nutrient treatments. First, differences in $J_s$ between radial depths of 5 mm and 15 mm were smaller for *R. mangle* and *A. germinans* in +P versus control and +N, drawing more $J_s$ into sapwood bands that would have been produced after fertilization (Figure 2). Second, +P initiated widely fluctuating $J_s$ for *A. germinans* at times, as exemplified in the very bottom graph of Figure 2. Control and + N had more consistent patterns of daily $J_s$ ramp-up and ramp-down. Third, despite no differences in maximum $J_s$ at a sapwood depth of 15 mm by species, treatment by radial depth interactions for control vs. +P treatments affected $J_s$ and gave rise to scaled changes in $S$.

### Stand water use ($S$)

All mangrove stands surveyed at DDNWR were in some phase of Hurricane Charley recovery. Average $S$ among the 85 basin forest

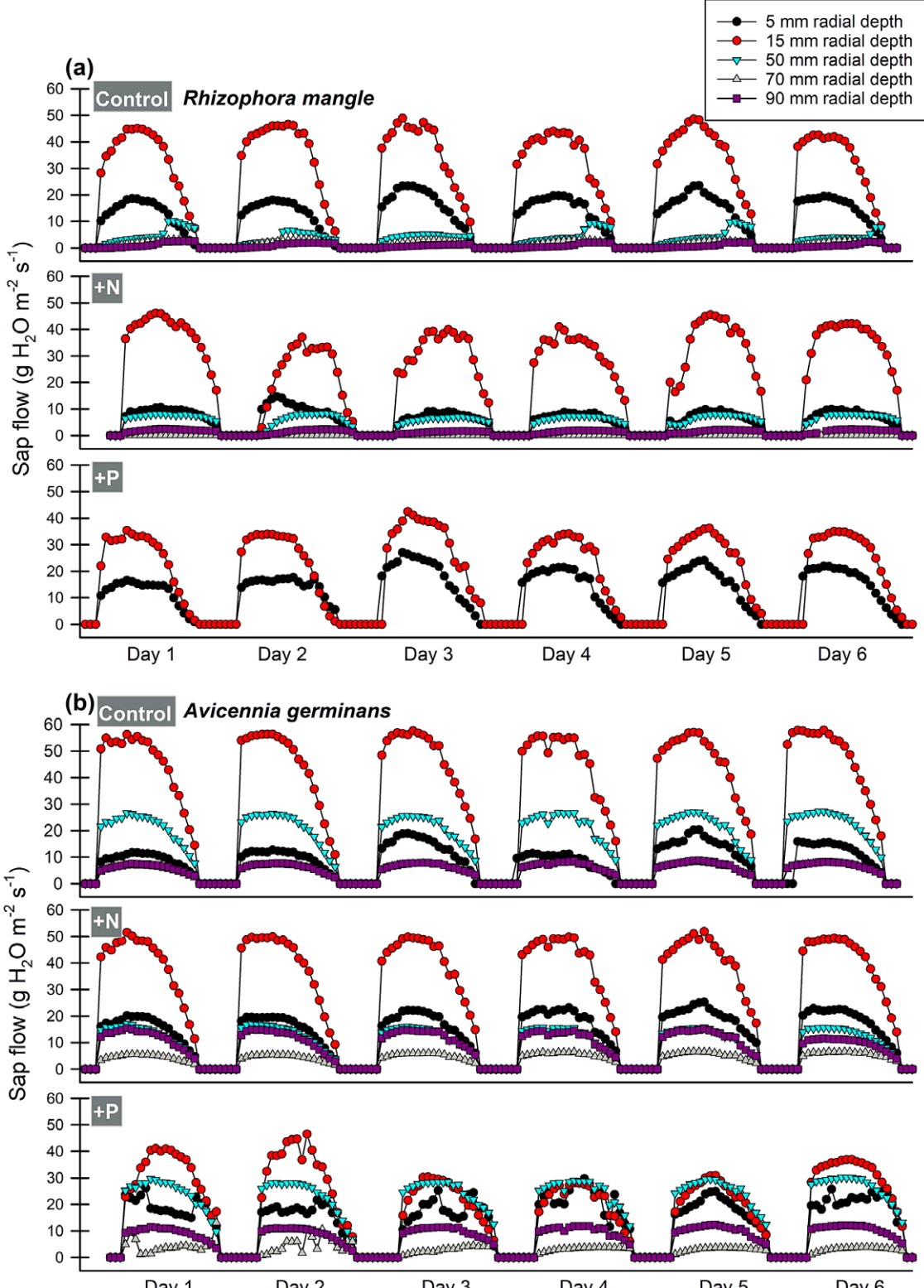

**Figure 2.** (a) Representative sap flow ($J_s$) response for the mangrove species, *R. mangle*, at radial sapwood depths of 5, 15, 50, 70 and/or 90 mm in control (unfertilized), nitrogen (+N) and phosphorus (+P) treatments at Ding Darling NWR. (b) Representative sap flow ($J_s$) response for the mangrove species, *A. germinans*, at radial sapwood depths of 5, 15, 50, 70 and/or 90 mm in control (unfertilized), nitrogen (+N) and phosphorus (+P) treatments at Ding Darling NWR. Annually, maximum rates of $J_s$ attenuated to 58% and 74% of maximum values in winter months (Dec/Jan) for *A. germinans* and *R. mangle*, respectively.

control plots was 594.3 ± 21.5 mm y$^{-1}$ (mean ± SE) (range, 226–1,181 mm y$^{-1}$) (Figure 3). Small plot sizes created greater variability in estimation of *S*, indicating that small forest structural shifts

among species and size classes affect this hydrologic metric. Likewise, average *S* among the 44 fringe control plots was 529.0 ± 27.2 mm H$_2$O y$^{-1}$ (range, 158–1,046 mm H$_2$O y$^{-1}$). Rates

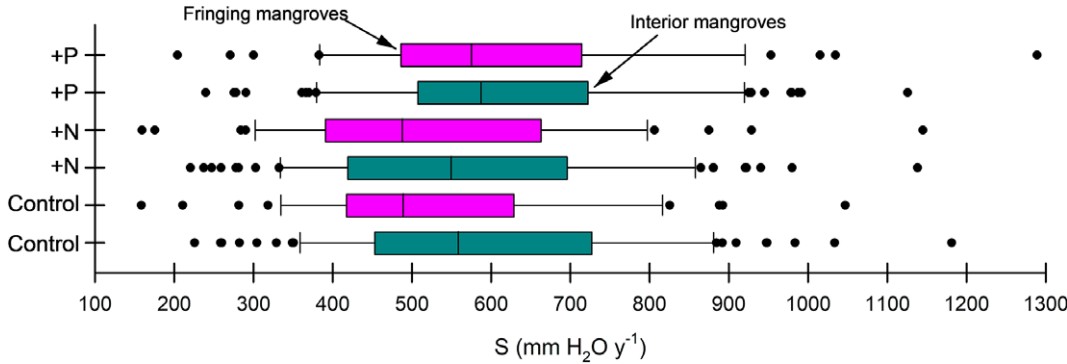

**Figure 3.** Median stand water use (*S*) for control (unfertilized), nitrogen-fertilized (+N) and phosphorus-fertilized (+P) fringe and basin mangrove forests simulated across 129 plots at Ding Darling NWR. Proceeding left to right on each plot, minimum (left bar), first quartile (Q1), median (Q2), third quartile (Q3) and maximum (right bar) values are depicted, with whiskers (•) highlighting data spread.

of *S* did not differ between zones at DDNWR for control (Kruskal-Wallis $H = 3.620$; $p = 0.057$), +N ($H = 1.630$; $p = 0.202$) or +P ($H = 0.158$; $p = 0.691$). Of note are the two lowest values of 226 mm $H_2O\,y^{-1}$ from a basin location and 158 mm $H_2O\,y^{-1}$ from a fringe location; several sites were sparsely populated with trees.

Stand water use was not influenced by the +N treatment versus control (contrast, $H = 0.405$; $p = 0.524$) but *S* was influenced by +P ($H = 3.867$; $p = 0.049$). Greater differences were revealed between +N and +P ($H = 6.310$; $p = 0.012$). Even slight changes in *S* among treatment combinations scale in ways to suggest important metabolic consequences brought on by increased phosphorus. *S* becomes $571.1 \pm 21.3$ mm $y^{-1}$ for +N and $615.7 \pm 20.3$ mm $y^{-1}$ for +P among basin mangroves, and points to a notable increase in water use when additional P is delivered. A similar increase in *S* within +P occurred in mangroves along the estuarine edge (fringe), where *S* becomes $609.2 \pm 32.0$ mm $y^{-1}$ (Figure 3). Overall, *S* remained static for +N treatments versus controls and was stimulated by 3.7–15.1% in +P-fertilized mangroves versus controls in basin and fringe mangroves, respectively. By individual species, the majority of increases in *S* under enhanced P loading was associated with *R. mangle* trees growing nearest the water's edge; greater individual tree water usage was compounded by tree size and reduced WUE$_i$ in *R. mangle*.

### Ecosystem carbon fluxes

#### Soil C burial and SSCA

Soil C burial ranged from 122.4 g C $m^{-2}\,y^{-1}$ to 185.7 g C $m^{-2}\,y^{-1}$ among basin sites by treatment, with a maximum difference between control and +P of 63.3 g C $m^{-2}\,y^{-1}$. Soil C burial rates were much lower for fringe than basin sites, ranging from 10.8 g C $m^{-2}\,y^{-1}$ to 34.2 g C $m^{-2}\,y^{-1}$ by treatment, with a maximum difference between control and +P of 23.4 g C $m^{-2}\,y^{-1}$. SSCA for basin mangroves was 35.7 g C $m^{-2}\,y^{-1}$ for control, 31.7 g C $m^{-2}\,y^{-1}$ for +N and 50.0 g C $m^{-2}\,y^{-1}$ for +P; however, SSCA was zero for all fringe plots.

#### Soil and pneumatophore CO₂ fluxes

$R_s$ differed between seasons ($F_{1,60} = 29.235$, $p < 0.001$), by sampling day within each season ($F_{2,60} = 10.607$, $p < 0.001$) and among treatments (+N, +P) within each season ($F_{2,60} = 7.568$, $p = 0.001$). $R_s$ from the control and +N were both significantly greater in the wet season compared to the dry season ($p < 0.05$) (Figure 4), increasing in the wet season by 23% for the control and 53% for +N versus the dry season. This suggests a higher sensitivity of soil

microbes to +N compared to +P during the wet season. However, during the dry season, $R_s$ from +P treatment was significantly greater than the control ($p = 0.045$) (Figure 4). This differs from the wet season results where +N, +P and control $R_s$ did not differentiate, demonstrating a shift in nutrient limitation between seasons. Furthermore, as soil temperature increased, $R_s$ tended to increase, particularly for +N. Soil temperature was positively correlated with moisture across all treatments ($0.64 \leq r \leq 0.80$; $p < 0.001$) (Table S1).

$R_p$ differed between seasons ($F_{1,67} = 54.736$, $p < 0.001$), by sampling day within each season ($F_{2,67} = 8.8345$, $p < 0.001$) and among treatments within each season ($F_{2,67} = 14.001$, $p < 0.001$). $R_p$ from the control and +P were both significantly greater in the wet season compared to the dry season ($p < 0.05$), increasing by 103% and 177% for control and +P in the wet season, respectively (Figure 4). During the dry season, $R_p$ from +N was significantly greater than the control ($p = 0.003$). This differed from the wet season. Furthermore, $R_p$ for +N did not differ between seasons and varied by only 5% versus the dry season. +P stimulated $R_p$ response more than +N between seasons, suggesting a higher sensitivity of live roots to +P. Thus, in contrast to $R_s$, as soil temperature increased, $R_p$ increased for control and +P treatments but not for +N. Warming of the soil drove greater $R_p$ versus $R_s$ once phosphorus was added (Table S1).

Annual $R_s$ and $R_p$ for the control summed to 754.7 g C $m^{-2}\,y^{-1}$ in 2019 and 751.5 g C $m^{-2}\,y^{-1}$ in 2020, with 10% lower $R_s$ than $R_p$ over both years (Figure 5). Summed $R_s$-generated and $R_p$-generated $CO_2$ fluxes ranged from $-3.6$ to 4.3 g C $m^{-2}\,d^{-1}$. Fluxes into the soil were uncommon but periodically documented during colder months. +N reduced summed $R_s$ and $R_p$ by 3.4% relative to the control, and +P increased summed $R_s$ and $R_p$ by 12.8% relative the control. Rainfall also differed between the 2 years, with cumulative depths of 983 mm in 2019 and 1,679 mm in 2020 (Figure 5).

#### Productivity and exchange of carbon

Canopy losses of $CO_2$ as $R_c$ averaged 475.1 g C $m^{-2}\,y^{-1}$ across both years, and were slightly higher for control, +N and +P in 2019 than in 2020, commensurate with the lower rainfall in 2019 (Figure 6). $R_c$ for +N and +P was 8.1% and 8.3% lower, respectively, than the control across both years, suggesting that $R_c$ differentiates little between +N and +P. In fact, for fringe mangroves, $R_c$ increased by 6% for +N versus controls. Annual uptake of C by the leaf canopy (or GPP) for the control was 2,665.1 g C $m^{-2}\,y^{-1}$ in 2019 and 2,592.4 g C $m^{-2}\,y^{-1}$ in 2020, ranging from 0.6 to 11.1 g C $m^{-2}\,d^{-1}$. GPP was 7.4% lower than control, or 2,435.1 g C $m^{-2}\,y^{-1}$, and

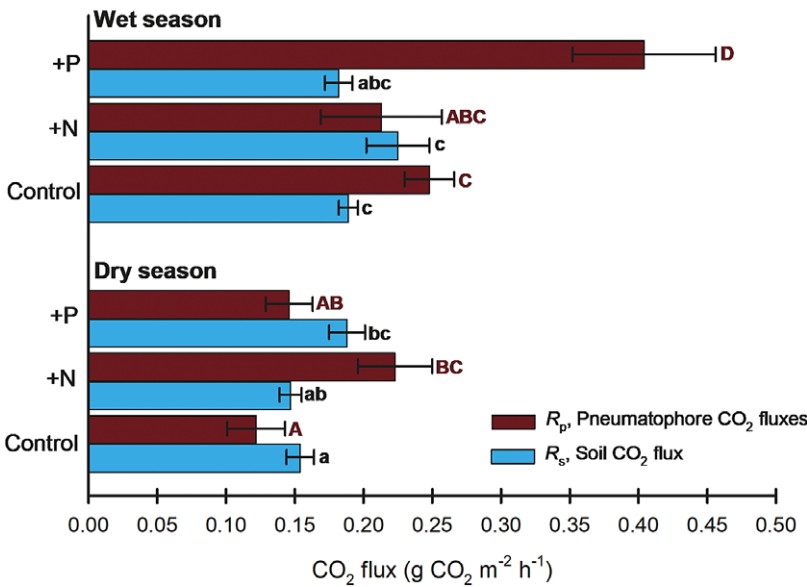

**Figure 4.** Soil $CO_2$ flux ($R_s$) and pneumatophore $CO_2$ flux ($R_p$) (mean ± SE) by sampling season for control (unfertilized), nitrogen-fertilized (+N) and phosphorus-fertilized (+P) mangrove forests at Ding Darling NWR.

**Figure 5.** Modeled soil $CO_2$ flux ($R_s$) and pneumatophore $CO_2$ flux ($R_p$) for control (unfertilized) mangrove plots at Ding Darling NWR for 2019 and 2020, as well as the annual timing and depth of rainfall for the two study years. While timing of $R_s + R_p$ for nitrogen-fertilized (+N) and phosphorus-fertilized (+P) simulations were similar to control (753 g C m$^{-2}$ y$^{-1}$), $R_s + R_p$ for +N was lower (727 g C m$^{-2}$ y$^{-1}$) and $R_s + R_p$ for +P was higher (850 g C m$^{-2}$ y$^{-1}$) across the two study years.

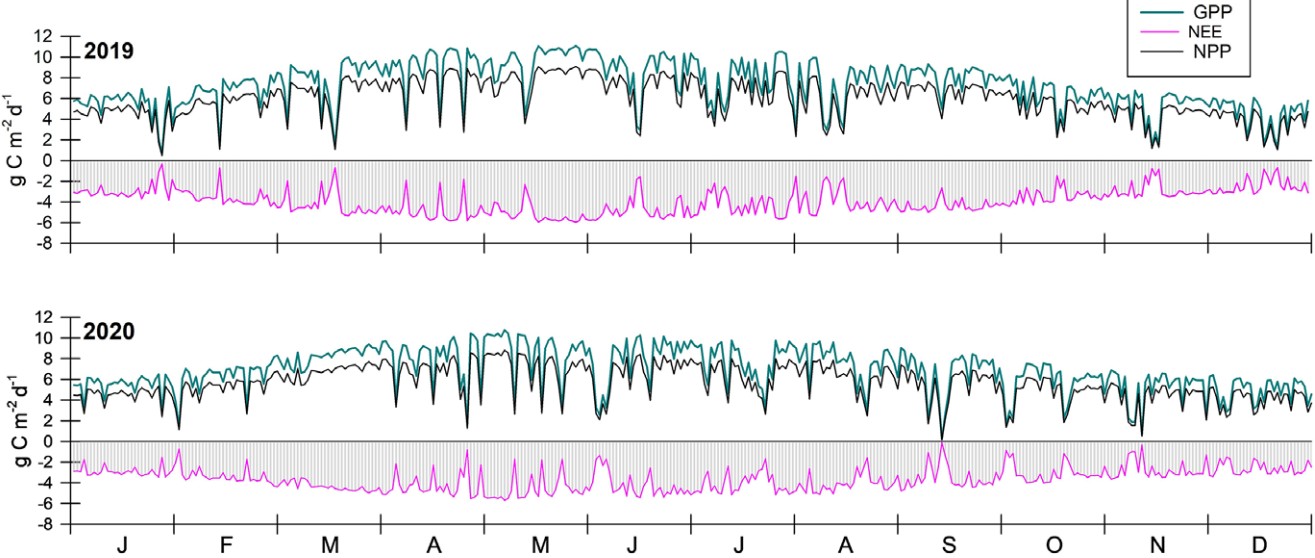

**Figure 6.** Modeled gross primary productivity (GPP), net primary productivity (NPP) and net ecosystem exchange (NEE) of carbon from control (unfertilized) mangrove plots at Ding Darling NWR for 2019 and 2020. While timing of GPP, NPP and NEE for nitrogen-fertilized (+N) and phosphorus-fertilized (+P) simulations were similar to control, relative rates differed significantly to affect scaling.

24.6% lower than control, or 1982.9 g C m$^{-2}$ y$^{-1}$, for +N and +P, respectively. GPP for control, +N and +P from basin mangroves averaged 2,756, 2,387 and 2,130 g C m$^{-2}$ y$^{-1}$ and from fringe mangroves averaged 2,383, 2,528 and 1,698 g C m$^{-2}$ y$^{-1}$. Subtracting losses of C as $R_c$ from GPP, average NPP becomes 2,245, 1,946 and 1,649 g C m$^{-2}$ y$^{-1}$ for control, +N and +P from basin mangroves over the two study years, and 1,977, 2,100 and 1,352 g C m$^{-2}$ y$^{-1}$ for fringe mangroves, respectively (Figure 6).

NEE averaged −1,382.3 g C m$^{-2}$ y$^{-1}$ for the control across both basin and fringe mangroves, ranging from −0.1 to −5.9 g C m$^{-2}$ d$^{-1}$ (negative values denote uptake of C) (Figure 6). The capacity for net atmospheric C fluxes into the basin forests was reduced by 18% for +N and 46% for +P (Figure 7), and for fringe forests was enhanced by 12% for +N and reduced by 59% for +P (Figure 8). For +N, NEE averaged −1,289 g C m$^{-2}$ y$^{-1}$ over the two study years across both hydrogeomorphic zones, but only −643 g C m$^{-2}$ y$^{-1}$ for +P, affecting the capacity for C export to support aquatic energy transformations. Lateral fluxes (all export) were estimated to range from 461 to 1,339 g C m$^{-2}$ y$^{-1}$ and was reduced considerably in +P treatments.

### Scaling the carbon budget to DDNWR

Scaling of GPP, NPP and NEE incorporates a mangrove area of 1,112 ± 116 ha at DDNWR (Peneva-Reed et al., 2021), comprised 68% basin and 32% fringe mangroves. The majority of the change in refuge-scaled C budgeting is associated with greater P loading (Table 3), experiencing less alteration by way of N loading alone. Specifically, additional P is expected to decrease GPP of DDNWR's mangroves by 7.2 Gg C annually, as well as causing reductions in NPP by 6.7 Gg C and NEE by half, to 7.7 Gg C. In addition, the system is likely operating at its maximum capacity for N uptake; GPP was not enhanced but rather was reduced refuge-wide by 2.3 Gg C when fertilized by N. $R_c$ losses scaled to around 4.9 Gg C y$^{-1}$ for both +N and + P, which were lower than control by approximately 0.4 Gg C y$^{-1}$ (Table 3). +N is projected to decrease lateral fluxes of C into estuarine waters by a small amount relative to controls, or by 1.4 Gg C y$^{-1}$ (Table 3). +P simulations suggest an

upending of DDNWR's mangrove C budget, stimulating higher stand water use, lower GPP, lower C uptake as NEE, higher $R_c$ and $R_p$ and reduced projected lateral export of C to estuarine waters by 7.3 Gg C annually versus controls.

### Discussion

#### Estimating productivity and lateral C fluxes

GPP among our six simulations ranged from 1,698 g C m$^{-2}$ y$^{-1}$ to 2,756 g C m$^{-2}$ y$^{-1}$, leaving an NPP of 1,352 g C m$^{-2}$ y$^{-1}$ to 2,245 g C m$^{-2}$ y$^{-1}$ once $R_c$ was subtracted. These estimations of GPP from DDNWR were similar to other mangrove locations in south Florida, ranging from 5.36 g C m$^{-2}$ d$^{-1}$ (or 1,956 g C m$^{-2}$ y$^{-1}$) to upwards of 9.6 g C m$^{-2}$ d$^{-1}$ (or 3,504 g C m$^{-2}$ y$^{-1}$) (Feagin et al., 2020), yet our technique was quite different. Sun et al. (2024) discovered that mangrove GPP averaged 2,054 g C m$^{-2}$ y$^{-1}$ (± 38.5 SE) globally. GPP was lowest at latitudinal limits of mangrove distribution (~1,000 g C m$^{-2}$ y$^{-1}$) and highest among a few hotspots along the equator (>3,000 g C m$^{-2}$ y$^{-1}$). GPP for southwest Florida's mangroves averaged 2,100 to >2,400 g C m$^{-2}$ y$^{-1}$ (Sun et al., 2024). Thus, a projected lowering of GPP by 656 g C m$^{-2}$ y$^{-1}$ in +P versus control simulation is an impactful decrease, and the primary reason why +P strongly affected NEE.

Unique among our study is the way we estimated lateral fluxes from DDNWR, including from our various treatments. We contend that difference calculations for lateral C flux estimation works as an approximation given that direct measurement of lateral fluxes lead to seasonal variation in flux rates as well and are extremely variable (e.g., Ohtsuka et al., 2020; Volta et al., 2020). Lateral fluxes are very difficult to assess directly and are costly. Nevertheless, we estimated that lateral C flux represented 52%, 58% and 71% of GPP for control, +N and +P from DDNWR among simulations and were most prominently affected in fringe forests, driven strongly by lower soil C burial (of 11–34 g C m$^{-2}$ y$^{-1}$).

Past C budgets for mangroves have identified a missing C component, or up to approximately 41–50% of GPP being unaccounted by total mangrove ecosystem C sequestration (Bouillon et al., 2008;

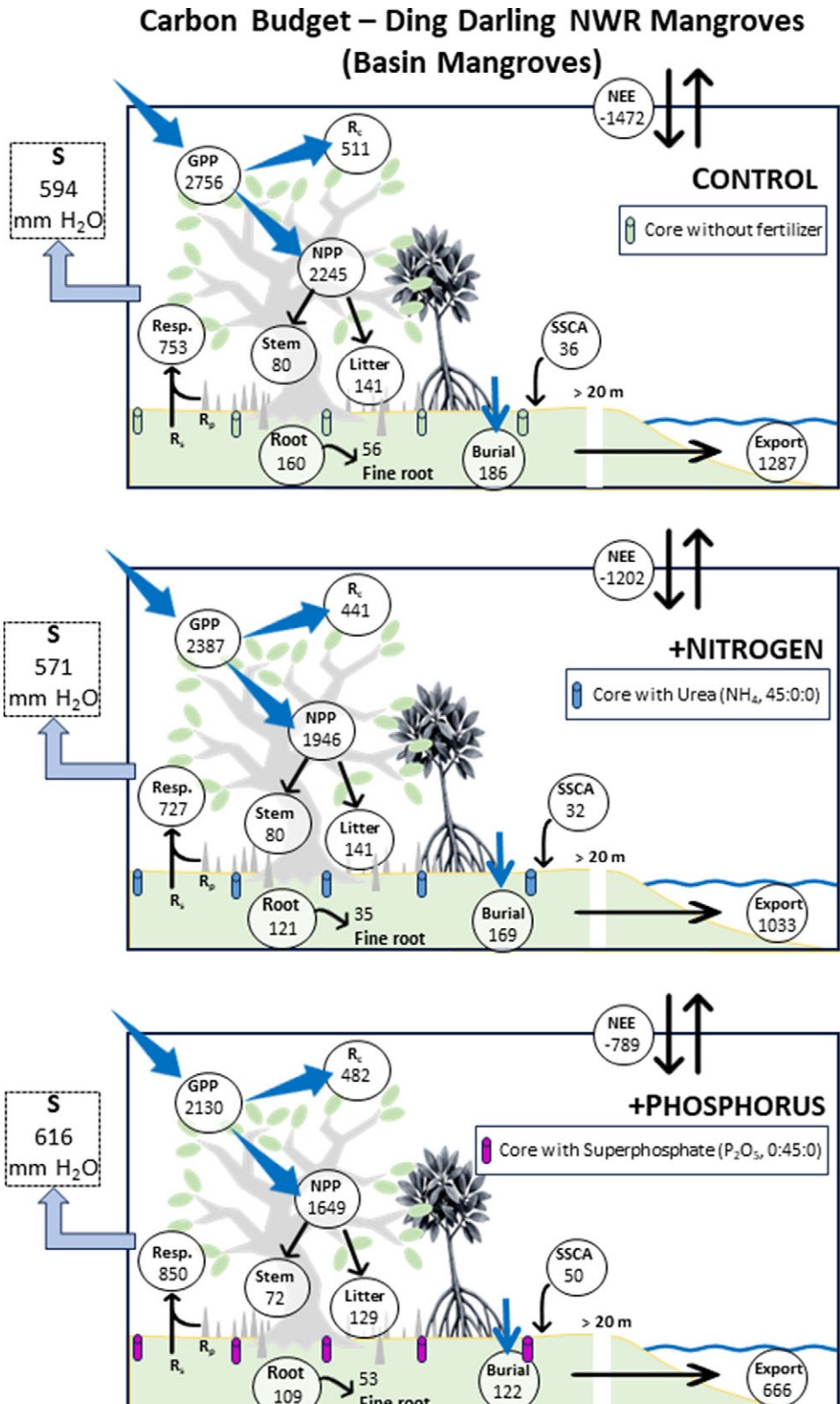

**Figure 7.** Annual carbon budgets (g C m$^{-2}$ y$^{-1}$) for basin mangrove forest versus water used (mm) for control (unfertilized), nitrogen-fertilized (+N) and phosphorus-fertilized (+P) mangroves at Ding Darling NWR. GPP = gross primary productivity, NPP = net primary productivity, NEE = net ecosystem exchange, $R_c$ = canopy respiration, Resp = soil ($R_s$) plus pneumatophore ($R_p$) CO$_2$ fluxes, SSCA = soil surface carbon accretion, Export = lateral C export and $S$ = stand water use. Litter, stem (allometry, Price et al., 2024), root (total) and fine root productivity from Conrad (2022). Negative values for NEE signify net uptake of C.

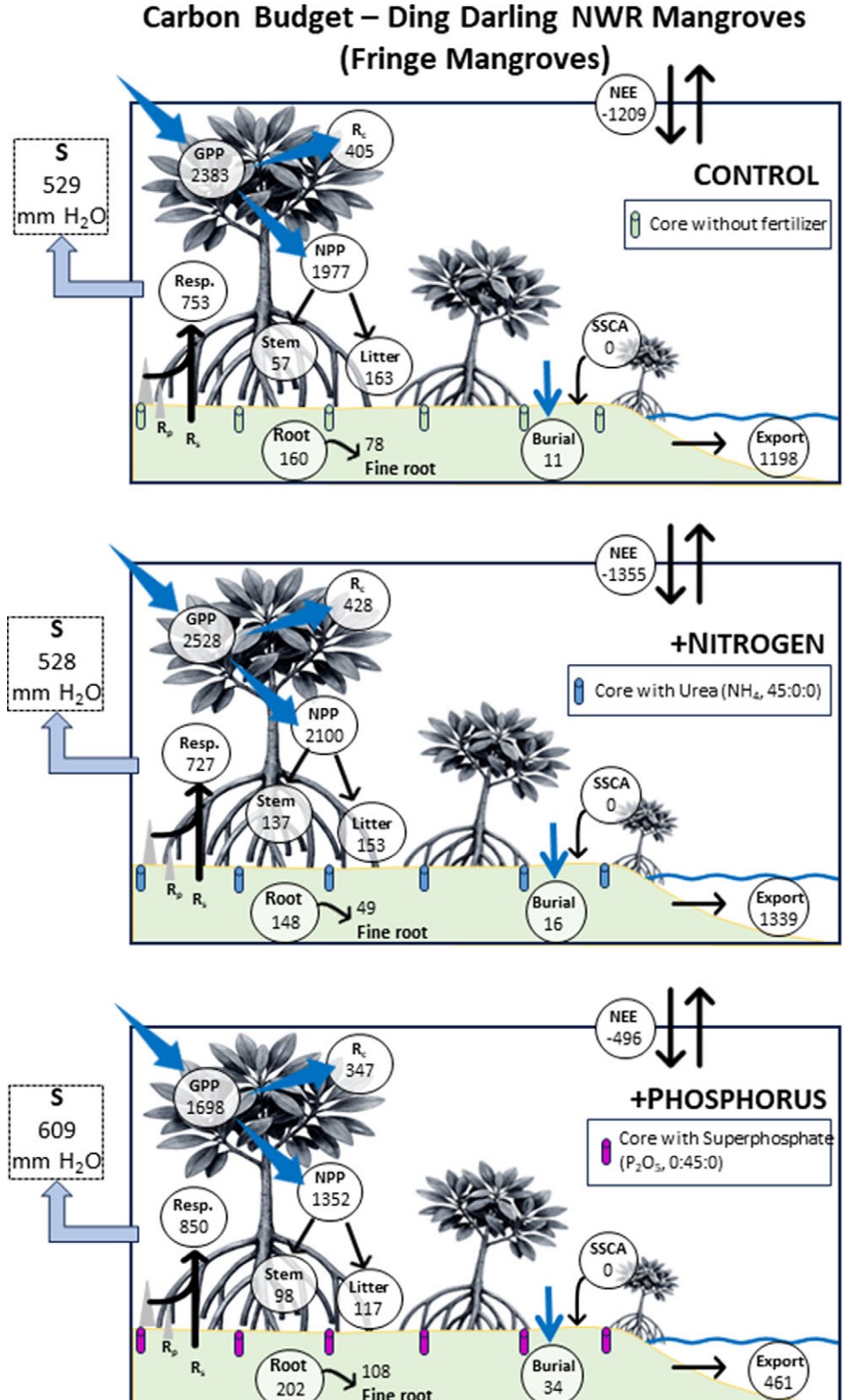

**Figure 8.** Annual carbon budgets (g C m$^{-2}$ y$^{-1}$) for fringe mangrove forests versus water used (mm) for control (unfertilized), nitrogen-fertilized (+N) and phosphorus-fertilized (+P) mangroves at Ding Darling NWR. GPP = gross primary productivity, NPP = net primary productivity, NEE = net ecosystem exchange, $R_c$ = canopy respiration, Resp = soil ($R_s$) plus pneumatophore ($R_p$) CO$_2$ fluxes, SSCA = soil surface carbon accretion, Export = lateral C export and $S$ = stand water use. Litter, stem (allometry, Price et al., 2024), root (total) and fine root productivity from Conrad (2022). Negative values for NEE signify net uptake of C.

**Table 3.** Scaled C budget projections by flux type for control, nitrogen (+N) and phosphorus (+P) based on a mangrove area of 1,112 ha (753 ha, basin; 359 ha, fringe) at Ding Darling NWR

| Carbon flux | Gg C y$^{-1}$ | | |
|---|---|---|---|
| | Control | +N | +P |
| GPP | 29.306 | 27.050 | 22.136 |
| $R_c$ | 5.305 | 4.859 | 4.871 |
| NPP | 24.001 | 22.191 | 17.265 |
| NEE | −15.424 | −13.920 | −7.716 |
| $R_s + R_p$ | 8.375 | 8.089 | 9.448 |
| SSCA | 0.269 | 0.239 | 0.376 |
| Soil C burial | 1.437 | 1.334 | 1.044 |
| C export | 13.987 | 12.587 | 6.673 |

Alongi, 2022). Alongi (2023) suggests, as we do here, that the majority of the missing C is received by estuarine waters as dissolved inorganic (DIC), dissolved organic (DOC) and particulate organic (POC) carbon. Tidal wetland lateral fluxes account for between 25% of total fixed carbon (Maher et al., 2018; Alongi, 2022) to as high as 80% (Najjar et al., 2018), falling within our range. We discovered that the addition of P to Sanibel's mangroves will likely press fluxes closer to 80% of total fixed carbon (Figure 7; Figure 8). While our analysis does simplify the estimation of lateral C fluxes and allows for treatment-specific dissection, seasonal distribution in lateral C fluxes would be sensitive to the timing of major Caloosahatchee River flows, which our technique does not consider. For example, lateral fluxes ranged from 1.56 g C m$^{-2}$ d$^{-1}$ in winter months to 4.21 g C m$^{-2}$ d$^{-1}$ in summer months from the mangrove-lined Fukido River, Japan (Ohtsuka et al., 2020).

### Application of S and WUE$_i$ toward GPP estimation

Our approach uses stand water use estimations to generate GPP among the 129 plots modeled, and this estimate had an average standard deviation of 196 mm y$^{-1}$. Given that 21–25 plots are required to determine a difference in stand water usage of just 200 mm y$^{-1}$ (Krauss et al., 2015), for example, modeling using our STrAP model or similar is the only real avenue for applying an S-to-GPP conversion at relevant spatial scales. Furthermore, WUE$_i$ values (Table S2) are sensitivity to relative humidity changes confounded by salinity concentration shifts across similar wet-to-arid environmental gradients (Clough and Sim, 1989). An option for future advancement might be the eco-physiological variable, $\partial E/\partial A$, which corresponds to the marginal water use of C gain (or $\lambda$; Liang et al., 2023). Among the different ways to represent water use strategy, using transpiration ($E$) relative to net CO$_2$ assimilation ($A$) as $\lambda$ was far superior to WUE$_i$ and intrinsic WUE (Liang et al., 2023). $\lambda$ is conservative across humidity gradients for wide application, such as for +N and +P fertilization. In time, standard values for $\lambda$ might be available by mangrove species and more easily incorporated into models. Estimates of S can be developed and scaled to GPP, NPP and/or NEE dependent on forest structural data and few other variables, such as $R_s$. The proof of concept toward forested wetland application is offered here.

Mangroves are known for maintaining relatively efficient water usage per C taken up at the tree- and stand-levels (Krauss et al., 2022), offering WUE$_i$ as a conservative trait in the taxa. Tang et al. (2006) suggests, as we do here, that water use efficiency can be used to estimate GPP, although hysteresis causes reduced predictability as vapor pressure deficits rise and PPFD confounds predictions with fluctuating vapor pressure deficits. In our model, we overcome this partially by limiting vapor pressure deficit to sap flow prediction to daily time periods only, such that time lags are averaged over the course of a day. This limits our confidence in reducing stand water usage to per-second or per-hour time scales; however, daily, weekly and annual time scales align well with published data and diurnal behavior from similar forest types (Krauss et al., 2015). We are confident in application to ≥ daily scales.

Finally, our exploration of model closure (Box 1, Supplementary Material) is encouraging considering that we did not include *R. mangle* prop root respiration, tree stem respiration or CH$_4$ fluxes when summing individual C productivity values in comparison to NPP. The average differential of 366 ± 26 g C m$^{-2}$ y$^{-1}$ (Table S3) between the two approaches may be explained by those missing components. Prop root respiration alone can contribute as much as 741 g C m$^{-2}$ y$^{-1}$ in CO$_2$ efflux (Golley et al., 1962; Troxler et al., 2015), although none of our experimental sites had dense prop roots. Likewise, soil and stem CH$_4$ fluxes can be as high as 4.9 to 5.3 g C m$^{-2}$ y$^{-1}$ (Qin et al., 2025), and stem CO$_2$ fluxes from trees in general can be upwards of 130 g C m$^{-2}$ y$^{-1}$ (Damesin et al., 2002).

### Nutrient-facilitated vulnerability of DDNWR'S mangroves

We discovered that additional loading of P will likely become increasingly detrimental to forest metabolism and biogeochemical energetics by reducing GPP and upsetting NEE from a strong atmospheric C sink to a weak C sink. Metabolically, +P stimulated greater stand water use caused by a reduced WUE$_i$, particularly in *R. mangle*, in the forest's struggle to balance C uptake against water usage (McDowell et al., 2022). Net C losses are also an indicator of tidal wetland degradation (Czapla et al., 2020a). The mangroves at DDNWR may be at the physiological limit of growth stimulation by nutrients, with limited C sink capacity without some rehabilitation. Indeed, soil surface elevation was reduced by +P (but not +N) over the three study years (Conrad et al., 2024), further implicating P as a concern if river loading increases. Malhotra et al. (2018) suggests that the presence of sulfur may affect crop yields in the presence of higher P, and the presence of seawater-borne sulfates in mangroves and high P concentrations is noteworthy. Excessive P loading on soil surfaces may also alter root morphology by promoting lateral expansion versus soil depth expansion and could result in decreased root biomass (Malhotra et al. 2018); however, in fringe environments root productivity was stimulated, particularly by fine roots with +P presumably in an effort mine additional N to balance N:P ratios. Root production was 32% lower in +P plots than controls for basin mangroves at DDNWR, but 26% higher in fringe mangroves, likely reflecting a consequence of higher soil P levels in basins (Conrad, 2022). Nevertheless, sewage pollution stunted the growth and led to mortality of *Avicennia marina* along the Red Sea, although a specific link to P was not made (Mandura, 1997). Excessive nutrients, particularly of N, may also increase susceptibility of mangroves during drought and high salinity events through greater proportional allocation to aboveground tissue versus roots (Lovelock et al., 2009).

The implied mechanism by our modeling implicates a water use vs. WUE$_i$ pathway that would drive decreased GPP with excessive phosphorus. For DDNWR, this was specifically expressed through *R. mangle*, which had a low water use efficiency of 0.009 g $CO_2$ (g $H_2O$)$^{-1}$ in +P versus that of the control, or 0.018 g $CO_2$ (g $H_2O$)$^{-1}$. WUE$_i$ for *A. germinans* was intermediate between control and +P, and our modeling assumed no change among plots for *L. racemosa*. That assumption would have only a modest influence unless WUE$_i$ in *L. racemosa* was very different among treatments. Overall, *R. mangle* contributed to 40.5%, *A. germinans* contributed to 35.4% and *L. racemosa* contributed to 24.1% of transpirational water loss across DDNWR over 2 years. Likewise, leaf and plant level WUE$_i$ decreased by 26% and 70%, respectively, in P-treated versus well-watered controls for the shrub, *Bauhinia faberi* (Song et al., 2010); however, reduced water use efficiency was not linked to reduced growth particularly when adequate water was supplied with P fertilization.

Eutrophication appears to take time to manifest observably in larger mangrove forests. After nearly a decade of fertilization, a salt marsh in Massachusetts, USA, contributed to salt marsh loss through an increase in aboveground biomass, increased decomposition and altered root biomass (Deegan et al., 2012). For DDNWR, reduced overall GPP appeared to cascade through the ecosystem with +P, reducing NPP, litterfall and soil and root $CO_2$ fluxes in basin (Figure 7) and fringe forests (Figure 8) but did not influence root growth consistently between zones. While mechanisms are not clear, we contend that ecosystem vulnerability is particularly acute in an estuary that is already eutrophic (DeGrove, 1981; Bricker et al., 1999; Lapointe and Bedford, 2007). Wetland ecosystem degradation by small degrees is overlooked and difficult to diagnose, echoing the ongoing legacy of the Florida Everglades (Douglas, 1997).

Metabolically, respiration from soil and belowground root structures was high at DDNWR (vs. Lovelock, 2008; Lovelock et al., 2014a; Hien et al., 2018), especially in the wet season when river water exchange is typically more direct. +N influenced $R_s$ more in the wet season and +P influenced $R_s$ more in the dry season, suggesting a potentially critical shift in bioavailable forms of N and P seasonally even with high total nutrient abundance. While this is puzzling, aboveground plant biomass, GPP and NEE were all enhanced by 1.5 years of N fertilization within a *Spartina alterniflora* salt marsh in North Carolina, manifesting more in erosive fringing marshes than within interior marshes linked to higher porewater sulfide concentrations (Czapla et al. 2020b). Returning to Deegan et al. (2012), continued fertilization eventually led to the demise of the aforementioned Massachusetts salt marsh, although the trajectory of change could not be determined over just the first 2 years of fertilization.

At DDNWR, it is likely that microbial and plant tissue N-stimulated (and some P-stimulated) responses are near saturation, and positioning across the refuge affects processes such soil oxygenation capacity. For example, P stimulation of $R_p$ fluxes was highest in the wet season, in contrast to $R_s$, indicating that a balance of pneumatophores versus open soil shifts across the refuge would influence respiratory losses. Soils and roots are responding differently, which also included the capacity for $CO_2$ fluxes into the soil for the $R_p$ term (as much as 0.28 g C m$^{-2}$ s$^{-1}$) but not as $R_s$. Other studies have documented $CO_2$ fluxes into soils of mangroves (Lovelock, 2008) and tidal freshwater swamps (Krauss and Whitbeck, 2012), while the capacity of mangrove pneumatophores for directing air downward is well-established

(Scholander et al., 1955). It is also possible that some seasonal N limitation occurs as P is depleted from root structures (e.g., P-fertilized fringe mangroves), and increased P loading from the Caloosahatchee River could eliminate further P uptake and degrade the role of DDNWR's mangroves in nutrient retention for enhanced water quality remediation of the estuary. It is important to note also that our fertilizer regiment of twice per year would differentiate from the repetitive, but inconsistent, pulsing delivered by future Caloosahatchee River flow. Phosphorus loading and other processes are driving persistent algae blooms in Pine Island Sound (Pearl et al., 2008), with the role of the mangroves remaining unclarified.

### Implications for managing the future of South Florida's mangroves

Phosphorus loading to Pine Island Sound and its surrounding mangroves will likely reduce lateral export of C from plots in the form of DOC, DIC and POC from a lower supply of fixed C, and make overall retention of C and nutrients less likely for the mangroves. However, given that the release of N and P from aquatic mineralization of C sources may currently be adding to eutrophication, +P might even partly offset aquatic eutrophication by reducing lateral fluxes while simultaneously reducing substrate capacity to support autotrophic to heterotrophic bioenergetic coupling. Such a conclusion would also have to be balanced against P-stimulated area losses to sea-level rise submergence that might occur more rapidly than in oligotrophic mangroves more characteristic of the Florida Everglades. One way to offset losses of elevation to resist submergence is through sediment deposition, which in organic C units ranged from 32 to 50 g C m$^{-2}$ y$^{-1}$ within DDNWR's basin mangroves but was absent in fringe mangroves. Even the basin mangrove rate is low compared to at least one salt marsh system, which registered 59 g C m$^{-2}$ y$^{-1}$ and 93 g C m$^{-2}$ y$^{-1}$ for controlled and N-fertilized plots (Czapla et al., 2020a), and is certainly not enough alone to keep DDNWR's mangroves persisting over the next century regardless of treatment applied.

Tidal wetlands must maintain capacity to build vertical surface elevations in order to migrate inland; they do not necessarily need to build faster than sea-level rise to do this (Conrad et al., 2024). While there may be limited concern from slow sea-level rise, land managers may work to improve habitat quality and build natural resiliency into the system to push back against modest accelerations in sea-level rise. Peneva-Reed et al. (2021) estimated total C standing stocks within the mangroves of DDNWR at 365 Gg C, with 288 Gg C belowground. Given a soil C burial rate of 1.437 Gg C y$^{-1}$ for control plots, and assuming that our control represents an historic condition for mangroves at DDNWR, it would take approximately 200 years to replace the soil C standing stock within the mangroves of DDNWR to a depth of 1 m. This assumption matches the contemporary basin mangrove accretion rate over the past 3-y of 0.64 cm y$^{-1}$ (Conrad et al., 2024, or 156 y to build 1-m of soil), the 50-y accretion rate of 0.57 cm y$^{-1}$ (Drexler, 2019, or 175 y to build 1-m of soil) and the 100-y accretion rate of 0.42 cm y$^{-1}$ (Drexler, 2019, or 238 y to build 1-m of soil). From this, we estimate turnover of the mangrove soil C resource at DDNWR to be 0.005, which suggests that 0.5% of all of the mangrove soil C at DDNWR turns over each year under current conditions (control). Long-term management is implied when sea-level rise is a concern, such that with a projected soil C burial shift to 1.044 Gg C y$^{-1}$ with loading of additional P, it would take approximately 275 years (e.g., 75 years

longer than control) to replace the soil C standing stock providing slower adjustment to rising seas.

Regulating P discharge down the Caloosahatchee River is among the options available to achieve the outcomes of inland mangrove migration potential, in-situ soil building, soil P retention and a more positive NEE and soil C burial balance for DDNWR's mangroves. Currently, soil P is being buried at exceptionally high rates of 36.6 g P m$^{-2}$ y$^{-1}$ within *R. mangle*–dominated fringe mangrove stands and 96.2 g P m$^{-2}$ y$^{-1}$ in basin mangroves (procedures outlined by Cormier et al., 2022), representing a substantial increase from average annual basin mangrove accumulation over the last 100 years within Sanibel Island's mangroves (1.25 g P m$^{-2}$ y$^{-1}$; Drexler, 2019). Mangroves may not be able to take up additional P without significantly altering patterns of surface elevation change or sedimentation, thus probably limiting further mangrove-facilitated water quality improvement aspirations for Pine Island Sound.

**Open peer review.** To view the open peer review materials for this article, please visit http://doi.org/10.1017/cft.2026.10025.

**Supplementary material.** The supplementary material for this article can be found at http://doi.org/10.1017/cft.2026.10025.

**Data availability statement.** All raw data collected during these studies are available to the public at ScienceBase.gov as follows: forest structure (Peneva-Reed and Zhu, 2019), soil cores (Drexler, 2019), sap flow (Duberstein et al., 2023), $R_c/R_p$ (Benscoter and Faron, 2023) and surface-elevation change (Conrad et al., 2023).

**Acknowledgements.** We thank Scott Covington (FWS), Paul Tritaik (FWS), Kevin Godsea (FWS), Erin Myers (FWS), Mark Danaher (FWS), Kurt Johnson (FWS), Avery Renshaw (FWS) and Nicole Cormier (Pontchartrain Conservancy) for project support, land management insight, field assistance and/or feedback on study design. Any use of trade, firm or product names is for descriptive purposes only and does not imply endorsement by the U.S. Government.

**Author contribution.** Conceptualization: K.W.K., J.R.C., J.A.D., Z.Z., I.C.F., B.W.B.; Literature review and research: K.W.K., J.R.C., E.J.W., J.Z.D., H.M., N.T.F., S.L.M., A.S.F., E.P.-R.; Writing and drafting: K.W.K.; Critical review and editing: J.R.C., J.A.D., E.J.W., J.Z.D., K.J.B., K.M.T.

**Financial support.** Funding was provided by the U.S. Geological Survey LandCarbon Program and U.S. Geological Survey Southeast Climate Adaptation Science Center.

**Competing interests.** The authors declare none.

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
