## [Reviewer Report]

General Comments

This manuscript investigates how experimental N and P fertilization affects carbon fluxes and ecosystem carbon balance in mangrove forests at J.N. “Ding” Darling National Wildlife Refuge, Sanibel Island, Florida. Over three years, the authors monitored sap flow, soil and pneumatophore CO₂ fluxes, and constructed carbon budgets to simulate future nutrient loading scenarios. Results suggest that additional phosphorus loading reduces productivity, reverses the net ecosystem carbon balance (NECB) from a sink to a source, and decreases lateral carbon export. These findings highlight the risk of eutrophication for mangrove resilience to sea-level rise.

Overall, I found very little critique for the MS. The science is sound, the article is very well structured and written.

I have only minor concerns:

The three-year fertilization and multi-metric monitoring is commendable. However, replication across only 18 plots raises concerns about spatial heterogeneity in mangroves; more detail on statistical power and limitations would help.

Nutrient Addition Protocol

The fertilizer treatments are clearly described, but I wondered whether applying nutrients twice annually adequately reflects the natural frequency and form of nutrient inputs from the Caloosahatchee River, which may occur in more irregular pulses and in different chemical forms. A short discussion of how these differences might affect plant and microbial responses would strengthen the interpretation.

Sap Flow → GPP Conversion

The use of sap flow and water-use efficiency (WUEi) to estimate gross primary productivity is innovative. At the same time, this approach involves several assumptions (e.g., that WUEi measured during a short campaign is representative year-round, and that scaling across species/size classes captures canopy heterogeneity). A more explicit discussion of uncertainties and how they propagate through the carbon budget would enhance confidence in the results.

Carbon Burial and Sedimentation

Soil carbon burial rates were estimated from pre-treatment cores and applied equally across treatments. While practical, this assumes burial is unaffected by nutrient additions, which may not always hold true. It would be valuable to comment on whether nutrient enrichment might alter burial processes (e.g., via decomposition or root production) over longer timescales.

Overall, I believe the paper makes a valuable contribution to: eutrophication, mangroves, carbon (BC) and management, spheres.

---

## [Reviewer Report]

Review: Cambridge Prisms: Coastal Futures

Title: Excessive phosphorus loading contributes to future vulnerability of mangrove ecosystems by reversing net ecosystem carbon balance

Authors: Ken W. Krauss and others

My comments are as follows:

1. The authors experimentally fertilised mangrove forests with nitrogen (+N; NH4) and phosphorus (+P; P2O5) for three years and monitored soil and pneumatophore CO2 fluxes, and tree sap flow in Avicennia germinans and Rhizophora mangle.

2. Individual tree and stand water use were modelled from which carbon budgets and other parameters were calculated for +N and +P vs. controls.

3. According to the authors, soil total P was 3-4 times higher on Sanibel Island than in other mangrove sites. Soil’s total N, however, was “not distinctive”.

4. The results indicated that added N altered turnover rates only minimally, while the addition of P led to a projected 0.48% loss of carbon per year. The authors suggest that P added to mangrove soils on carbonate settings leads to greater growth. Please add that others in several countries (including some authors of this manuscript) have also shown that adding N as well as N+P leads to increased growth and biomass.

5. The authors report no differences in N among sites, despite receiving added N fertilisers, which is surprising. P concentrations, however, were 3-4 times higher, despite the high inherent sediment P. Generally, mangrove forests are low-nutrient environments due to the infertility of upland soils and low terrigenous input. It is well established that nitrogen availability is one of the primary factors limiting the growth of mangroves. Please comment on this discrepancy in the results.

6. Granular nitrogen (urea NH4 – 45:0:0, N-P-K) or phosphorus (superphosphate P2O5 – 0:45:0, N-P-K) was used based on previous application experience (Feller, 1995; McKee et al., 2002; Feller et al., 2003; McKee et al., 2007). There were 16 fertilisation holes per plot. Holes were augured to a 2.5 cm diameter and 30 cm depth, and 150 g of fertiliser was added to each with replacement of soil. Fertilisers were applied twice annually for three years.

7. In this study, there is a serious problem with the above method of fertiliser application. In these field experiments, fertiliser was applied by inserting it into holes in the soil and then resealing the holes with soil. In an agricultural setting, this method of fertiliser application is appropriate. This experiment, however, was conducted in an intertidal area with diurnal flooding. The applied fertilisers, which are water-soluble, would have been washed away within a few days of application. There is no comment on tidal influence in the experimental sites.

8. Were soil concentrations of N and P determined in the fertilised compared to adjacent unfertilized sites? There is no mention of this in the manuscript.

9. The authors need to address the points raised above before acceptance of this manuscript.

10. In addition to the points listed above, the manuscript is well-written and presented.

---

## [Editor Report]

Dear authors

Both reviewers give useful input where further explanation / detail is required on methods and data interpretation. Please address these, we look forward to the revised article.

---

## [Reviewer Report]

Concentrations of P and N in the sediment after fertilisation are now mentioned in the supplementary material. It is still surprising that trees did not respond to added N.

---

## [Editor Report]

Thank-you for the revised manuscript that is now ready for publication.

Both reviewers were happy with the revised version and changes made.